# Nesting Biology and Ecology of a Resin Bee, *Megachile cephalotes* (Megachilidae: Hymenoptera)

**DOI:** 10.3390/insects13111058

**Published:** 2022-11-16

**Authors:** Waseem Akram, Asif Sajjad, Hamed A. Ghramh, Mudssar Ali, Khalid Ali Khan

**Affiliations:** 1Department of Entomology, Faculty of Agriculture and Environment, The Islamia University of Bahawalpur, Bahawalpur 63100, Punjab, Pakistan; 2Research Center for Advanced Materials Science (RCAMS), King Khalid University, P.O. Box 9004, Abha 61413, Saudi Arabia; 3Unit of Bee Research and Honey Production, King Khalid University, P.O. Box 9004, Abha 61413, Saudi Arabia; 4Biology Department Faculty of Science, King Khalid University, P.O. Box 9004, Abha 61413, Saudi Arabia; 5Institute of Plant Protection, Muhammad Nawaz Shareef University of Agriculture, Multan 60000, Punjab, Pakistan; 6Applied College, King Khalid University, P.O. Box 9004, Abha 61413, Saudi Arabia

**Keywords:** nesting, biology, ecology, *Megachile cephalotes*, resin bee

## Abstract

**Simple Summary:**

*Megachile cephalotes* is a solitary bee that is widely distributed in Sindh and Punjab, Pakistan. It has been reported as an effective pollinator of *Grewia asiatica* and some other crops. Bees are declining around the world, threatening the productivity of field crops, vegetables, and fruits. Among the Megachilini tribe, the only European leaf-cutting bees, *Megachile rotundata* has been artificially reared and conserved on a commercial scale in different parts of the world. Some recent studies have shown a high pollination potential of *M. cephalotes* owing to its gregarious nesting and foraging behavior. In the present study, the nesting biology and ecology of *M. cephalotes* were reported for the first time in this region. The bees remained active during the spring and summer seasons, and females preferred to construct their nests in bamboo sticks and wooden blocks. They used plant resin for the construction of brood cells and placed several types of pollen grains in these cells. The males took fewer days to become adults than the females. The present study will help in the commercial-scale artificial nesting and conservation of *M. cephalotes*.

**Abstract:**

We report the nesting biology and ecology of *Megachile cephalotes* Smith, 1853 for the first time in Pakistan. Wooden and bamboo trap nests were deployed at three different locations in Bahawalpur district, Pakistan, from January 2020 to May 2021. A total of 242 nests of *M. cephalotes* were occupied in all three locations with the maximum abundance in the Cholistan Institute of Desert Studies. *Megachile cephalotes* remained active from March to September (the spring and summer seasons). In a nest, females made 7–8 brood cells each having a length of 1.2–2.3 cm. Plant resin was used to construct cells and mud or animal dung to plug the nest entrance. A vestibular cell was also made between the outermost brood cell and the nest entrance that ranged from 1.4 to 2.5 cm in length. No intercalary cells were observed in the nests. The males took 65.3 days to become adults, while the females took 74.78 days. The sex ratio was significantly biased toward females in all three locations. *Grewia asiatica* was the predominant pollen grain species found in the brood cells. *Megachile cephalotes* were observed collecting resin from *Acacia nilotica*, *Prosopis juliflora*, and *Moringa oleifera*. Three cleptoparasites of this species were also recorded: *Euaspis carbonaria*, *Coelioxys* sp., and *Anthrax* sp. This study set up a background to encourage new studies on artificial nesting and provides tools for proper biodiversity management and conservation.

## 1. Introduction

Managed bees are considered the most efficient pollinators throughout the world, but wild bees have also received considerable attention for the last few decades due to their high pollination efficiency and because they provide equivalent services to those of managed bees [1,2]. Sometimes wild bees also enhance the pollination services provided by managed bees through their behaviors [3,4]. Among 20,000 species of bees, the majority of the species are wild [5]. The population of bees is declining throughout the world, ultimately threatening the productivity of major crops, vegetables, and fruits [6,7]. Several factors have been attributed to bee decline, e.g., habitat degradation, climate change, intensive use of pesticides, predators, and parasites [8,9,10]. The availability of suitable foraging resources and nesting habitats helps minimize the decline [11,12,13,14].

The family Megachilidae, including more than 4000 species, occurs throughout the world [15,16]. Within Megachilidae, *Megachile* Latreille, 1802 is the most diverse genus, comprised of 32 subgenera and 431 species only in the neotropical region, along with several unidentified species [16,17,18,19,20]. Numerous species of *Megachile* are efficient pollinators [21,22,23,24,25,26,27] and only *M. rotundata* could be artificially reared and conserved on a commercial basis [28,29]. In Pakistan, 18 species of the genus *Megachile* are known [15], yet no information is available on the nesting biology and ecology of megachilid bees.

*Megachile* species are solitary and highly adaptive and build their nests in pre-existing cavities, e.g., wooden logs, hollow stems of bamboo and roses, burrows in the soil, cracks and crevices, and slits in rocks or manmade structures [16,30,31,32,33]. For the construction of their brood cells, female *Megachile* spp. use a variety of materials, e.g., leaf pieces, flower petals, mud, pebbles, and a combination of resin and salivary material [16,30,34,35,36,37,38,39,40]. A recent study reported the use of plastics from wrappings, flags, and bags [41]. This behavior applies to some species, but it is not a general behavior of all *Megachile* species. Poor knowledge about nesting biology is the main barrier to utilizing the diverse megachilid bees as a management tool for pollination services.

Each nest consists of a linear series of brood cells with pollen provisions (nectar and pollen mixture) packed by the female [42]. The female lays eggs on pollen provisions and finally closes the cell with pieces of leaves [43]. Once the first cell is fully completed, the females repeat this process several times from the closed to the open end of the nesting cavity. When the nest is filled with cells, they close the nest from the outside with masticated leaves or mud to protect their offspring [42]. Depending on their resources, *Megachile* bees can be oligolectic [44] or polylectic [16,45]. Most research has been conducted on the nesting biology of leafcutter bees [46,47,48,49,50,51,52,53], while resin bees are poorly understood [42,54,55,56,57].

By using bee hotels or trap nests, the nesting biology and ecology of bees are studied for various purposes, i.e., the study of nest architecture [50,58,59], natural history [60,61,62,63], evolution [16], crop pollinators [64], population monitoring and community structure [13,65], bioindication and recording changes in habitat type [66,67,68] and as a tool in conservation [59,69] and quantitative ecology [70,71,72]. The latter includes the quantification of multiple trophic interactions between bees, wasps, their food objects, and natural enemies [73].

*Megachile* (*Callomegachile*) *cephalotes* is a solitary bee that is widely distributed in Sindh and Punjab, Pakistan [15]. From Pakistan, it has been reported as an effective pollinator of *Grewia asiatica* [25,26]. The genus *Grewia* has 140 to 150 species, of which only *G. asiatica* is of commercial importance as a fruit crop in subtropical and tropical regions [74,75]. Little is known about the nesting biology and ecology of this bee [54]. The aim of this study was to provide information about the nesting biology (nest architecture and pollen types used) and ecology (seasonality and plant species providing pollen and nectar) of *M. cephalotes* for the first time in Pakistan. This nesting behavior differs from that observed in other parts of its range and can provide the basis for comparative studies.

## 2. Material and Method

### 2.1. Study Site

The study was conducted from January 2020 to May 2021 at three different sites in Bahawalpur district: Cholistan Institute of Desert Studies (CIDS; 29.3784° N, 71.7696° E), the Agricultural Research Farm (ARF; 29.3714° N, 71.7652° E), and the Fisheries Complex (FC; 29.3863° N, 71.6300° E). CIDS is comprised of a highly diverse landscape of over 40 hectares, including desert, orchards, unmanaged land, lawns, horticultural landscaping, and some agricultural land. A diverse array of floral resources remains available year-round. Natural or artificial nesting resources, i.e., cracks or holes in mud walls, hollow tree branches, and bamboo or reed sheds are also abundant. The ARF is 65 hectares and is comprised of agricultural and horticultural crops. Cotton and maize are the major crops, and the need-based application of insecticides is a usual practice. The FC is 27 hectares and is comprised of *G. asiatica* fields, citrus orchards, and some other horticultural flowering plants. Natural and artificial nesting resources are also abundant for bees, i.e., cracks or holes in brick walls, empty fish ponds, and hollow tree trunks or branches.

The climate of the district is arid with mild winters and hot summers. There are four seasons in this zone: spring (March to May), summer (June to September), autumn (October to November), and winter (December to February) [76]. The mean daily minimum and maximum temperatures are 28 °C and 42 °C in the summer and 6 °C and 22 °C in the winter, respectively. The average annual rainfall in Bahawalpur is 169.8 mm [77].

### 2.2. Nesting Material

At each site, one wooden frame with dimensions of 152 cm (length) × 92 cm (width) × 30 cm (depth) was installed on 3 January 2020. Each wooden frame consisted of seven partitions, and each partition was filled with certain nesting material. The nesting materials included wooden logs, mud blocks, wooden blocks, cardboard tubes, bamboo sticks, dry reeds, and plastic straws. Two wooden plates (18 cm long) with longitudinal groves were stacked on each other so that they made complete holes [53]. Five such plates were stacked and tied with adhesive tape to form a wooden block. Only bamboo sticks and wooden blocks were considered for further study, as the females of *Megachile cephalotes* constructed their nests in bamboo sticks and wooden blocks. One thousand bamboo sticks and one hundred wooden blocks were placed in each wooden frame.

### 2.3. Nest Sampling

Trap nests were checked weekly for occupied nests. The occupied nests were removed and replaced with new ones. The collected nests were marked with the nest collection date, and the open end was covered with nylon mesh bags to check the adult emergence. All the collected nests were placed in cages, and the emergence of adult *M. cephalotes* was observed. The date of emergence of male and female individuals was recorded. The time period between nest collection and the emergence of individuals was considered the developmental period. The sex ratio of *M. cephalotes* was also noted.

### 2.4. Nesting Biology and Ecology

Five completely occupied wooden blocks were shifted to the laboratory and dissected to study the nest structure. The following parameters were evaluated: total nest length, nest diameter, brood cell length, vestibular cell length, total number of brood cells/nest, and nest closure material. The digital Vernier caliper was used to measure the length and diameter of the nests. For comparison, four occupied nests in bamboo sticks were X-ray-photographed with an X-ray machine (Figure 1).

The floral host plants were recorded by directly observing *M. cephalotes* foraging at the study sites. The weekly random walks were made between 9:00 am to 5:00 pm to record the plant species foraged by *M. cephalotes*. The resin collection was also directly observed in the field by focusing on resin-producing plants. To identify and measure the pollen grains of the brood cells, reference glass slides were made first by removing pollen grains from the available blooming plants during the spring and summer seasons when *M. cephalotes* were active. These vouchers aided in the identification of pollen provisions in brood cells. Pollen provisions were sampled from 15 different brood cells, acetolized, and mounted on glass slides [78,79]. Using the reference slides, pollen grains collected from the brood cells were identified. From each sample, 1000 pollen grains were identified and counted under a 60× stereomicroscope.

### 2.5. Statistical Analysis

A *t*-test was applied to compare male and female *M. cephalotes* in terms of the mean number of days to become adults. The Chi-square goodness of fit test was applied to determine the effect of location on the sex ratio of *M. cephalotes*.

## 3. Results

A total of 242 nests (bamboo sticks = 184, wooden blocks = 58) of *M. cephalotes* were collected from January 2020 to December 2020 at three locations: CIDS, ARF, and FC. The maximum number of nests was collected from CIDS, followed by the FC and ARF (Figure 2).

The first occupied nest was seen on 26 March 2020. The males of *M. cephalotes* emerged earlier than the females. The emergence started in the last week of April with a sharp increase until July and then a gradual decline until September. We did not observe any emergence from October 2020 to February 2021. These five months represented the hibernation period of *M. cephalotes*. The emergence of these hibernated populations started again in the first week of March 2021. This showed that *M. cephalotes* remained active from March to September, comprising the spring and summer seasons (Figure 3 and Figure 4).

For the study of nest architecture, five completely occupied wooden block nests were collected in May 2020. These nests were similar in length and diameter. The number of cells ranged from 7 to 8 with minimum and maximum cell lengths of 1.2 and 2.3 cm, respectively (Table 1). In each nest, all the cells were constructed in a linear series and aligned horizontally. The construction of the first cell started at the base of the nest (Figure 5). Before constructing the first cell, female *M. cephalotes* collected resin and deposited it at the base of the nest, making a thick layer. The brood cells were slightly rounded at the base and elongated. The female made several trips to collect pollen and nectar to provision the brood cells in the form of pollen lobes at the cell base. After provisioning the first cell, the female laid a single egg on the provision mass. After oviposition, the female made more trips to collect resin, which was used for cell closure. They made seven to eight such cells in a single nest. They also left a space (a vestibular cell) ranging from 1.4 to 2.5 cm long between the outermost brood cell and the nest entrance (Figure 1). The nests were then plugged with mud or animal dung at the entrance (Figure 5).

There was a statistically significant difference in the development period of male and female *M. cephalotes*. The males became adults earlier than the females, i.e., after 65.30 and 74.78 days, respectively (Table 2). There was no impact of the three locations on the sex ratio of *M. cephalotes*. The sex ratio of *M. cephalotes* was significantly biased toward females at all three locations (Table 3).

*Grewia asiatica* was the predominant pollen grain species found in the brood cells of *M. cephalotes*, followed by *Alhagi graecorum* and *Parkinsonia aculeata*. Among all plant species, the minimum number of pollen grains was found for *Rosa indica* (Table 4).

The floral host plants of *M. cephalotes* are presented in Table 5. Females of *M. cephalotes* visited six plant species. The maximum visits were recorded on *G. asiatica*, followed by *A. graecorum*, *P. aculeata*, and *Cajanus cajan* (Table 5). Resin-producing plants visited by *M. cephalotes* are presented in Table 6.

### Parasitoids

Adult parasitoids of three species emerged from 242 nests of *M. cephalotes*. Two belonged to the order Hymenoptera, i.e., *Euaspis carbonaria* (Megachilidae) (Figure 6a) and *Coelioxys* sp. (Megachilidae) (Figure 6b), and one belonged to Diptera, i.e., *Anthrax* sp. (Bombyliidae) (Figure 6c).

## 4. Discussion

We reported the nesting biology and ecology of *Megachile cephalotes* for the first time in Pakistan. In the present study, the maximum number of nests was collected from CIDS, followed by the FC and ARF. CIDS consists of a natural semi-desert landscape with abundant nesting cavities (cracks or holes in mud walls, hollow tree branches, and bamboo or reed sheds) and floral resources. The FC is mostly covered with *G. asiatica* fields that provide adequate nectar and pollen for bees. High species richness and an abundance of flowers usually favor the species richness and abundance of bees [80,81]. Apparently, there are factors other than floral availability that can limit wild bees. Empirical evidence shows that nesting resources affect the abundance of bees. There is a need to study how the availability of natural nesting resources affects solitary bee populations [82,83,84] since this resource is also essential for bee existence.

The results of the present study showed that *M. cephalotes* remained active from March to September, comprising the spring and summer seasons. The data over several years suggest that solitary bees exhibit marked spatiotemporal fluctuations in their abundance and diversity [16]. The species of the Megachilini tribe are reported to have two generations a year in northwestern India, and emergence occurs from late February or early March until May and again at the start of August until November [85]. Kunjwal et al. [86] reported that in India, *M. cephalotes* is multivoltine by nature and remains active from March to December. Moreover, they recorded the peak activity of *Megachile* spp. two times throughout the year: from March to May and from October to November. Rauf et al. [14] also found that *M. cephalotes* remained active from March to November in Punjab, Pakistan. Kumari and Kumar [85] reported gregarious nesting and foraging behavior in *M. cephalotes* that could be helpful for pollination. Depending on the locality and resources, Megachile bees remain active during the hottest months of the year [16]. Several seasonal and regional factors might affect the voltinism in Megachile bees. For example, *M. rotundata* is univoltine in its native range in Eurasia but bi- or multivoltine in North America, where it was accidentally introduced in the late 1940s [87,88,89,90]. Hence, there is a need to study these factors thoroughly in order to determine their effects on voltinism.

In the present study, female *M. cephalotes* construct their nests in bamboo sticks and wooden blocks with lengths and diameters of 15–16 cm and 7 mm, respectively. This species preferred to construct its nests in bamboo sticks with diameters ranging from 8 to 10 mm [14] and lengths from 8.2 to 18 cm [54]. In our study, females of *M. cephalotes* constructed 7 to 8 brood cells with minimum and maximum cell lengths of 1.2 and 2.3 cm, respectively. Previously, the nesting biology of this species had been described in India [54]. They reported that females construct 7 to 12 brood cells in castor sticks with minimum and maximum cell lengths of 1.0 and 1.8 cm, respectively. The number of brood cells constructed can vary depending on the sex ratio, nest length, and age of the female provisioner. Cavity nesters typically provision multiple nests. As a female ages and approaches mortality, cavity nesters tend to build fewer cells per nest since if she dies before completing the nest closure, the cells will be unprotected.

In the current study, the vestibular cell was recorded in each nest with lengths ranging from 1.4 to 2.5 cm. Many studies have reported that the majority of *Megachile* species made a vestibular cell in each nest [50,53,57,91,92]. In the present study, female *M. cephalotes* constructed their brood cells exclusively using plant resin. Contrarily, Gupta et al. [54] found that this bee solely used mud for its nest construction. Species of the subgenus *Callomegachile* mostly collect plant resin but sometimes also collect mud for their nest construction, hence the name resin bees [93]. Plant resin is a versatile material that can easily be shaped when fresh and structurally rigid when hardened, and it can be used as a nest substrate or to bind loose structural or camouflaging materials for the construction of brood cells. Resin is also waterproof, which permits moisture regulation in the nest, and some resins display potent anti-microbial properties [94,95,96,97,98,99].

The sex ratio of *M. cephalotes* was significantly biased toward females at all three locations. Our results are in agreement with those of Torretta et al. [91], who found that the sex ratio in *M. gomphrenoides* was female-biased. Contrarily, few studies have found that the sex ratio of *Megachile* spp. is male-biased [51,100]. Nest length, availability of floral resources, and flight distance from the nest to floral resources are the major factors that affect the sex ratio of bees. In short nesting cavities, the sex ratio shifted toward the sex whose brood cells were closest to the nest opening [101]. Gruber et al. [102] found that lengths shorter than 15 cm favored male production, which acted as a shield for females from parasites [103]. A female-biased sex ratio is the outcome of rich floral resources, whereas a male-biased sex ratio is the outcome of low floral resources [104]. Peterson and Roitberg [105] found that more sons were produced with an increase in the flight distance between the nest and floral resources. Furthermore, females provide fewer resources, which also results in fewer offspring.

*Grewia asiatica* was the predominant pollen grain species found in the brood cells of *M. cephalotes*, followed by *Alhagi graecorum* and *Parkinsonia aculeata*. *Megachile cephalotes* exhibited a wider range of plant interactions [106]. The yellow flowers of *G. asiatica* are zygomorphic with a good “side on” advertisement [107]. Bees tend to forage on food resources near their nest site. Bees exhibit floral constancy as a strategy that targets rewards and balances energy expenditure, i.e., they tend to feed on the most dominant species in the landscape [108]. Megachilid bees prefer zygomorphic and yellow-colored flowers, i.e., flowers with high UV reflection and pigmentation patterns and with a “side on” advertisement [109,110,111,112].

In the present study, adult parasitoids of three species, *Euaspis carbonaria*, *Coelioxys* sp., and *Anthrax* sp. emerged from the nests of *M. cephalotes*. Similarly, Rauf et al. [14] found that the nests of *M. cephalotes* were parasitized by *E. carbonaria* and *Anthrax* sp. All three species have already been reported as cleptoparasites of *Megachile* spp. in different regions [113,114,115,116,117,118].

## 5. Conclusions

In the present study, we reported the nesting biology and ecology of *M. cephalotes* for the first time in the arid zone of Punjab, Pakistan. *Megachile cephalotes* pursued their nesting activity in the spring and summer seasons (March to September) and hibernated in the autumn and winter seasons (October to February). Bamboo sticks and wooden blocks were the preferred nesting materials of *M. cephalotes*. Females constructed 7 to 8 brood cells in a single nest with a male-to-female brood ratio of 1:2.8. Males developed into adults earlier than females. *Grewia asiatica* was the major host plant for adults and broods. Future studies should investigate the effects of ecological and regional conditions on the voltinism of this bee and develop commercial rearing methods for crop pollination.

## Figures and Tables

**Figure 1 insects-13-01058-f001:**
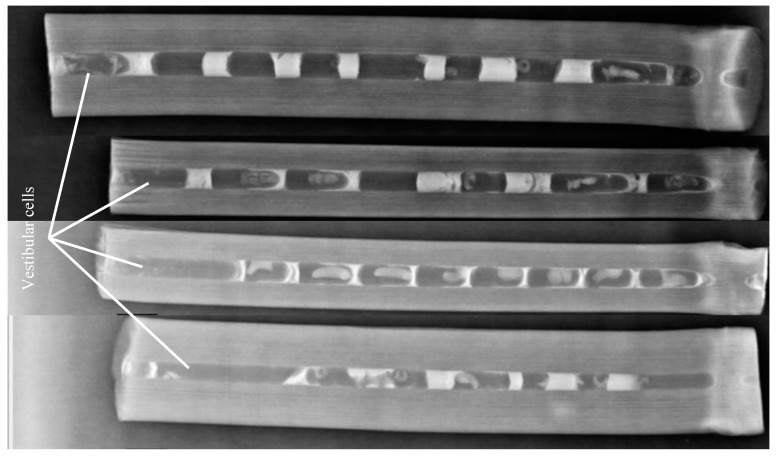
X-ray photographs showing four nests of *Megachile cephalotes*. Nest entrance is at the left.

**Figure 2 insects-13-01058-f002:**
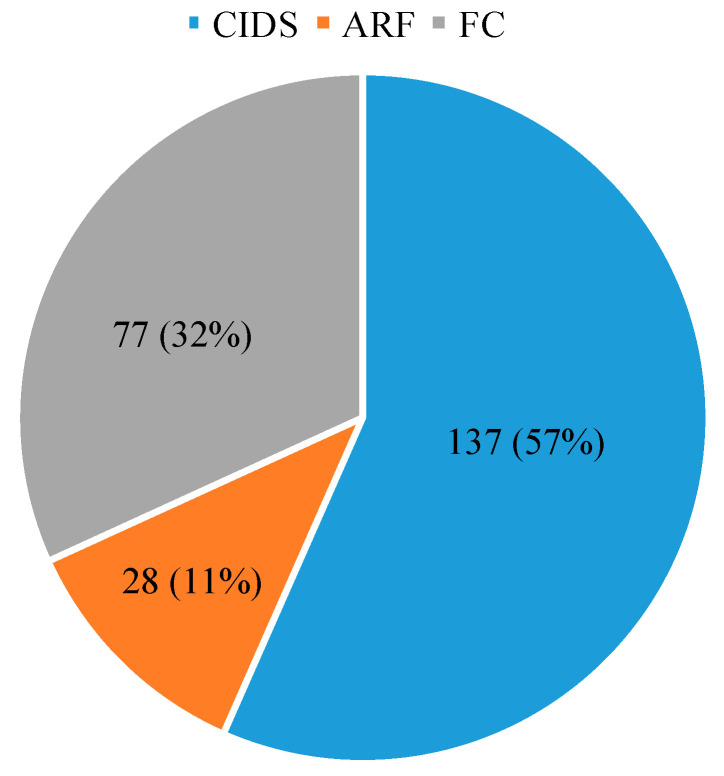
Number of occupied nests of *Megachile cephalotes* at three different sites from January to December 2020.

**Figure 3 insects-13-01058-f003:**
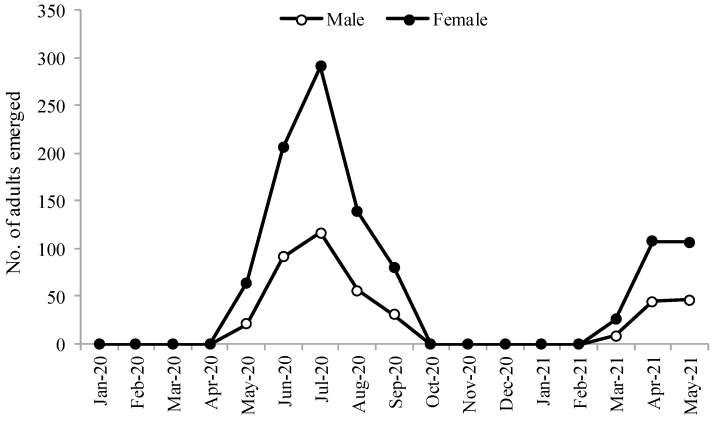
Emergence of male and female *Megachile cephalotes* from January 2020 to May 2021.

**Figure 4 insects-13-01058-f004:**
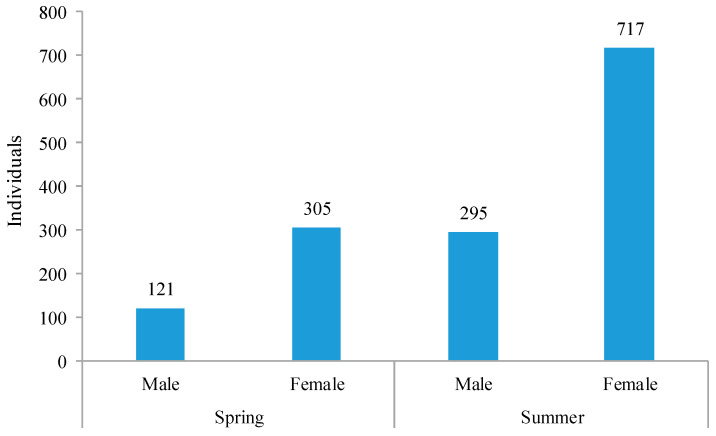
Total emergence of male and female *Megachile cephalotes* in spring and summer seasons from January 2020 to May 2021.

**Figure 5 insects-13-01058-f005:**
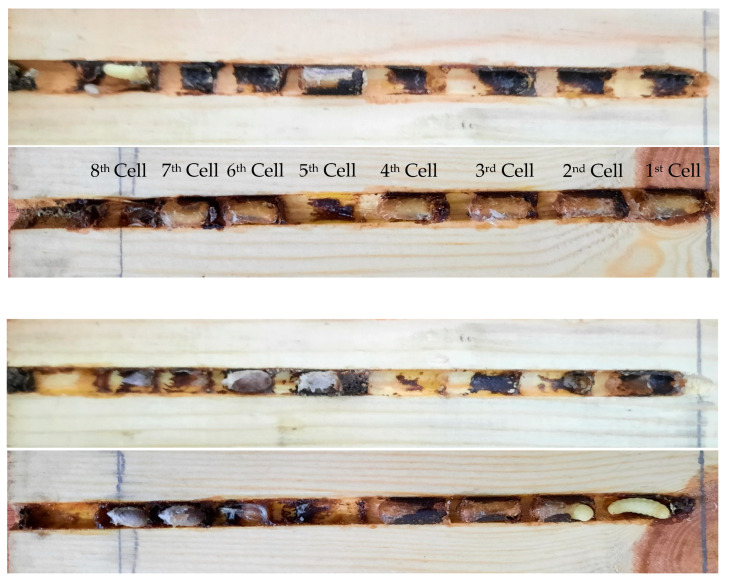
Samples of wooden plates consisted of *Megachile cephalotes* larval, pre-pupal, and pupal stages.

**Figure 6 insects-13-01058-f006:**
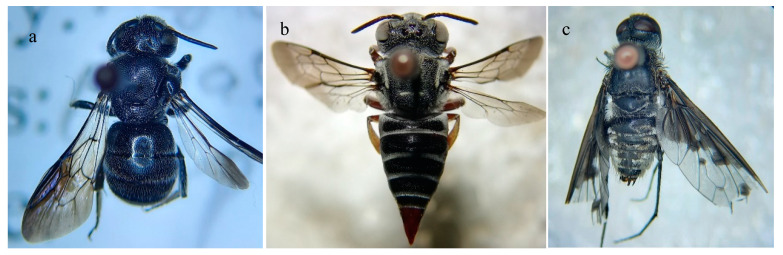
Parasitoids of *Megachile cephalotes* (**a**) *Euaspis carbonaria*, (**b**) *Coelioxys* sp., and (**c**) *Anthrax* sp.

**Table 1 insects-13-01058-t001:** Nest measurements of *Megachile cephalotes*-occupied nests.

	Nest 1	Nest 2	Nest 3	Nest 4	Nest 5
Nest length (cm)	16	15	15.5	16	16
Nest diameter (mm)	7	7	7	7	7
Nest closure	Mud	Mud	Animal dung	Animal dung	Mud
Number of cells	8	8	7	8	7
Vestibular cell length (cm)	1.5	2	2	1.4	2.5
Cell length (cm)					
1st	1.7	1.8	1.65	2.1	1.8
2nd	1.2	1.6	1.7	1.5	1.65
3rd	1.9	1.5	1.7	1.7	1.3
4th	1.9	1.8	1.5	1.5	1.5
5th	1.7	2	2.3	1.6	1.6
6th	1.3	1.3	1.5	1.4	1.4
7th	1.25	1.2	1.3	1.6	1.4
8th	1.6	1.2	-	1.6	-

**Table 2 insects-13-01058-t002:** Mean development days required for male and female *Megachile cephalotes*.

Sex	Means ± SE	t-Critical	t-Critical	df	*p*-Value
**Male**	65.30 ± 3.67	1.9803	2.0936	118	0.0384
**Female**	74.78 ± 2.69				

**Table 3 insects-13-01058-t003:** Sex ratio of *Megachile cephalotes* that emerged from nests in CIDS, ERF, and FC.

Locations	M:F	Sex Ratio	Chi-Square	*p*-Value
**CIDS**	254:568	1:2.0	119.946	<0.0001
**ERF**	44:121	1:2.8	35.933	<0.0001
**FC**	118:333	1:2.8	102.494	<0.0001

**Table 4 insects-13-01058-t004:** Percentage of pollen types used by *Megachile cephalotes*.

No. of Brood Cells	*Alhagi graecorum*	*Grewia asiatica*	*Parkinsonia aculeata*	*Rosa indica*	*Prosopis juliflora*	*Cajanus cajan*
1	31.22	63.75	1.94	-	3.09	-
2	-	98.67	-	-	1.33	-
3	65.54	21.87	-	-	-	12.59
4	-	19.44	79.05	-	1.51	-
5	-	-	98.91	0.45	0.64	-
6	12.76	82.11	5.13	-	-	-
7	72.12	8.47	11.65	1.17	-	6.59
8	0.67	99.33	-	-	-	-
9	12.68	86.91	0.35	0.06	-	-
10	92.39	6.81	-	0.8	-	-
11	36.19	62.91	0.75	-	-	0.15
12	15.97	84.03	-	-	-	-
13	84.91	-	-	-	7.33	7.76
14	-	97.05	-	0.25	-	2.7
15	10.11	-	89.73	-	-	0.16

**Table 5 insects-13-01058-t005:** Floral host plants visited by *Megachile cephalotes*.

Plants	Family	Habit	Flower Type and Color	Floral Reward	Abundance of *M. cephalotes*	Percent Proportion
*Alhagi graecorum*	Fabaceae	H, p	Z, pi	N, P	21	22.58
*Grewia asiatica*	Malvaceae	S, d	A, y	N, P	47	50.54
*Parkinsonia aculeata*	Fabaceae	T, p	A, y	N, P	12	12.9
*Rosa indica*	Rosaceae	d	A, w	N	1	1.08
*Prosopis juliflora*	Fabaceae	S, e, p	A, y	N, P	3	3.23
*Cajanus cajan*	Fabaceae	S, p	Z, y	N, P	9	9.68

Column 3: H = herb, S = shrub, T = tree, p = perennial, d = deciduous e = evergreen; Column 4: Z = zygomorphic, A = actinomorphic, pi = pink, y = yellow, w = white; Column 5: N = nectar, P = pollen.

**Table 6 insects-13-01058-t006:** Resin-producing plants visited by *Megachile cephalotes*.

Plants	Family	Habit
*Acacia nilotica*	Fabaceae	T, E
*Prosopis juliflora*	Fabaceae	S, P
*Moringa oleifera*	Moringaceae	T, P, D

Column 3: T = tree, S = shrub, E = evergreen, P = perennial, D = deciduous.

## Data Availability

Available on demand.

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
