# Peer review of "Nesting Biology and Ecology of a Resin Bee, Megachile cephalotes (Megachilidae: Hymenoptera)"

_insects, 2022, doi:10.3390/insects13111058_

Round 1

Reviewer 1 Report

The manuscript by entitled "Nesting biology and ecology of a resin bee, Megachile cephalotes (Megachilidae: Hymenoptera)" by Akram et al., describes the nesting biology and ecology of one of the important pollinator. The authors have very effectively portrayed the methodology and the outcomes of their research. The manuscript is very well written, it has some minor mistakes that I have highlighted on the annotated file for improvement.

Thanks

Reviewer 2 Report

The paper provides a description of the nesting biology and ecology of a resin bee, Megachile cephalotes.

works like this have long been underestimated as beeing scarcely innovative, with the result of neglecting the biology of these insects so crucial for the maintenance of biodiversity in every part of the planet, and not having enough information for their conservation nowadays that would be needed. Therefore, works that, like this, specify the biology of each individual bee species should be encouraged.

This particular article is sufficiently detailed and presents interesting aspects of this species, although the methods may be better detailed and the correspondence between methods and results can be improved. However, further investigation could have been conducted with the available materials, and therefore it is a bit of a missed opportunity. On the other hand, this species of Megachile has been shown to accept artificial nests with ease and therefore, through a more accurate survey of the state of the art on the observations conducted on artificial nests of wild bees, further research will lead to knowing further aspects of this species. For further comments, please look at Specific Comments as following.

SPECIFIC COMMENTS

72 This behavior applies to some species, it is not a general behavior of all species of the genus Megachile

78 polylectic;

95-96 Please, emphasize here that Grewia asiatica is a crop;

123, 127 Also use the metric system here, as in all the rest of the paper;

131 How many bamboo sticks and how many wooden blocks? See comment on line 173;

136 Wire house, what is meant?;

173 More than the absolute number of nests collected in each locality, it is interesting the percentage of nests occupied on the available nests;

176-183 The description of the adults activity requires major revision. There is confusion of terminology between emergence, activity and occurrence. E.g. There can be no emergency from June to September if the species is univoltine. A female emerges (makes her first flight) only once, and from that moment she begins her activity (mating, search for a suitable tunnel for nesting, supply, oviposition ...) which can be detected through its occurrence in front of the entrance hole of the tunnel chosen as nest. All these phases can be carefully studied if each female and the corresponding tunnel are marked with appropriate colors. Since this has not been done, it is necessary to better describe in the methods how individuals are counted. The graphic and the text describing it do not match. The emergence started in the last week of May, but from the graph it seems instead in mid-April. A decrease in occurrences from mid-July to the end of September is not sudden, it is instead gradual;

208 Methods for observing the developmental period of males and females are not described in the Methods section. Please provide an accurate description;

249-250 The meaning of the sentence is not understood; 

261-262 The term megachilid bees includes Osmiini and Anthidiini; Kunijwal 2016c says: "Megachile bees flies during the warmest parts of the year, with 2-4 generations per year" (not megachilid!) And cites Michener 2007 for this statement, but I could not find this attestation (2 to 4 generations ) in Michener 2007. Please check carefully, as most Megachile species probably only have one generation per year;

280-281 In the present study, female M. cephalotes construct their brood cells by exclusively using plant resin. " In the rest of the paper it is said that they also use animal dung;

328 Future studies. My advice is to indicate that future studies should include the method of marking females

Reviewer 3 Report

The manuscript "Nesting biology and ecology of a resin bee, Megachile cephalotes (Megachilidae: Hymenoptera)" reports on the nesting biology of M. cephalotes in Pakistan. This paper provided information on this bee's regional natural history, which is useful for future studies or management. While this is not the first study on the nesting biology of this species, it is interesting that the biology differs from other parts of its range. The potential environmental or evolutionary causes of these differences could be addressed to broaden the scope of this paper. Regardless, this paper provides useful background about this species' nesting habit, which benefits the field.

Introduction: A line could be added that this nesting behavior differs from that observed in other parts of its range and can provide the basis for comparative studies.

Line 123: Change to metric.

Line 175: Males are reported to emerge first, but Figure 3 has males and females emerging during the same months. So what time frame is meant by males emerging first? Provide empirical evidence.

Line 181: I think the parenthetical (Fig. 2, 3) is actually referring to (Fig. 3, 4) as there is no reference to figure 4 in the results section, and a reference here would be apt.

Table 4, 5: It is interesting there is Rosaceae sp. pollen found in the pollen balls, but they were not recorded as a floral host, but Calotropis procera was identified as a floral host, yet no pollen in the pollen ball. Why do the authors suggest that is?

Line 235: Do the authors have any numeric data on the rate of parasitoids? How many nests are affected, and is one parasitoid more prevalent than another?
